# Caffeine, alcohol, khat, and tobacco use during pregnancy in Butajira, South Central Ethiopia

**Alehegn Aderaw Alamneh[1]\*, Bilal Shikur Endris[2], Seifu Hagos Gebreyesus[2]**

**1** Department of Human Nutrition and Food Science, College of Health Sciences, Debre Markos University, Debre Markos, Ethiopia, **2** Departement of Nutrition and Dietetics, School of Public Health, Addis Ababa University, Addis Ababa, Ethiopia

\* alehegn12aderaw@gmail.com

**Data Availability Statement:** All relevant data are within the paper and its Supporting Information files.

**Funding:** The author(s) received no specific funding for this work.

## Abstract

### Background

The use of excessive caffeine and consumption of alcohol, cigarette, and khat during pregnancy can result in adverse health effects on the fetus. The World Health Organization (WHO) recommends a daily caffeine intake not exceeding 300 mg. Likewise, pregnant women are recommended to avoid alcohol, khat and tobacco use. However, the prevalence's of the use of substances among pregnant women were not well studied in developing countries such as Ethiopia. Therefore, the study aimed to estimate the prevalence of caffeine and alcohol consumption, khat chewing, and tobacco use during pregnancy and identify key factors associated with excess caffeine consumption.

### Methods

We conducted a community based cross-sectional study and used a random sampling technique to recruit 352 pregnant women. We adapted a questionnaire from Caffeine Consumption Questionnaire-Revised (CCQ-R), Alcohol Use Disorder Identification Test (AUDIT), Global Adult Tobacco Survey (GATS), and Ethiopian Demographic Health Survey 2016 for caffeine, alcohol consumption, tobacco use, and khat chewing assessment, respectively. We conducted non-consecutive two days 24-hour recall to determine the habitual intake of caffeine from caffeinated beverages and foods. Prevalence with 95% confidence interval was estimated for excess caffeine intake per day, alcohol consumption, khat chewing, and passive tobacco smoking. We ran a multivariable binary logistic regression model to identify factors associated with excess caffeine intake.

### Results

Almost all pregnant women (98.2%) consumed caffeine as estimated using the 2 days 24-hour average. The median daily caffeine intake was 170.5 mg and ranged from 0.00 mg to 549.8 mg per day. In addition, 17.6% (95% CI: 13.9%, 22.0%) of them had a daily caffeine consumption of 300 mg and above exceeding the WHO recommended daily caffeine intake

**Competing interests:** The authors have declared that no competing interests exist.

during pregnancy. The prevalence of alcohol consumption and Khat chewing were 10.0% (95% CI: 7.2%, 13.7%) and 35.8% (95% CI: 30.8, 41.0%) respectively. None of the pregnant women were active tobacco smokers. However, 23.2% (95% CI: 19.0, 28.0%) were passive tobacco smokers. We found that pregnant women in the richest wealth quintile (AOR = 3.66; 95% CI: 1.13, 11.88), and the first trimester of pregnancy (AOR = 4.04; 95% CI: 1.26, 13.05) had higher odds of consuming excessive caffeine.

## Conclusions

The study showed a considerable magnitude of substance use among pregnant women in the study area. Given this findings, we recommend, programs and services focusing on pregnant women to consider addressing substance use.

## Introduction

Substance use is defined as the inappropriate consumption of medicines, drugs, or other materials including prescription drugs, over-the-counter drugs, street drugs, alcohol and tobacco [1]. Caffeine is a stimulant substance found in coffee, tea, cocoa (chocolate), and kola nuts (cola), soft drinks, energy drinks, and some over-the-counter medications [2]. Coffee is one of the most popular consumed beverages in the world and the most common sources of high caffeine [2].

Even though caffeine contains several chemical components that may provide health benefit in reducing dementia [3], insulin resistance, type 2 diabetes mellitus [4, 5], Parkinson disease [5], cirrhosis and advanced hepatic fibrosis [6–11], excess intake is not recommended; especially during pregnancy [12]. This is for the fact that caffeine can cross the placenta into the amniotic fluid and fetus and results in adverse pregnancy outcomes. The American Pregnancy Association and March of Dimes recommends that a pregnant woman should not take more than 200mg caffeine per day, which is around 355 milliliters coffee [13]. In 2003, Experts in Canada conducted a systematic review to evaluate the effects of caffeine on human health and recommended that a reproductive age woman should not take 300 mg and above caffeine per day. In addition, the World Health Organization (WHO) recommends that daily caffeine intake should not exceed 300 mg during pregnancy [12, 14].

Alcoholic beverages are drinks containing ethyl alcohol or ethanol which is an intoxicating ingredient. Alcohol is a central nervous system depressant and can cross the placenta. Therefore, since there is no safe amount of alcohol, a pregnant woman is advised to avoid drinking alcohol during pregnancy [15].

Globally, 9.8% of women consume alcohol while they are pregnant [16]. In the Eastern Africa WHO region, the estimated prevalence of alcohol consumption during pregnancy among the general population ranged from 3.4% in Seychelles to 20.5% in Uganda. In Ethiopia, the magnitude of alcohol consumption during pregnancy ranged from 7.9% to 34% [17, 18].

Alcohol consumed during pregnancy is the leading preventable cause of developmental disabilities and birth defects. According to the World Health Organization report, 1 in 100 babies is estimated to be born with alcohol-related damage [19]. One of the problems is Fetal Alcohol Spectrum Disorder (FASD), which is an umbrella term that covers all alcohol-related diagnoses [20].Besides, heavy alcohol consumption during pregnancy increases the risks of low birth

weight, preterm birth [21], small for gestational age [22] and childhood leukemia in young children [23].

Globally, 8% of women aged 15 years and above were tobacco smoker. The tobacco epidemics continue to shift from high-income countries to low- and middle-income countries, with a recent increase in the prevalence of tobacco smoking among women, which is expected to rise to 20% by 2025 [24]. Based on the 2011and 2016 Ethiopian Demographic Health Survey (EDHS), the overall prevalence of tobacco use among reproductive age women was 0.8% [25, 26]. Different chemicals from cigarette smoking impair the structure and function of the placenta [27]. Therefore, smoking during pregnancy has been associated with preterm birth, restricted fetal growth and low birth weight [28, 29], and this leads to a higher risk of childhood obesity and non-communicable diseases in later life [30, 31].

Khat refers to the leaves and the young shoots of the plant *Catha edulis Forsk*, a species belonging to the plant family Celastraceae. Khat contains many different compounds such as cathinone [32]. As a result, khat chewing during pregnancy may reduce placental blood flow [33]. The WHO Expert Committee on Drug Dependence (ECDD) critical review result showed that khat chewing during pregnancy may have different obstetric effects like low birth weight, stillbirths, and impaired lactation [32]. In 2008, a community-based cross-sectional study in Yemen showed that about 40.7% of women reported chewing khat while pregnant [34].

Different factors were identified as risk factors for substances use during pregnancy. Younger age, white ethnicity, not being religious, low socio-economic status, being nonimmigrant, performing less frequent antenatal consultation, null parity, husband alcohol consumption, previous history of alcohol and other illicit drugs use, unplanned pregnancy, lack of awareness about the harmful effects of alcohol on the fetus and peer pressure were the identified factors associated with alcohol consumption [35–38]. Being divorced, unemployed, younger age, low educational level and low socio-economic status, living with smoker, criminal history, working in receipt of social services, alcohol and illicit drug use, being fair to poor in perceived health, being previous heavy smoker, having at least one chronic disease and mental illness, and not having a regular medical doctor were the identified factors associated with cigarette smoking during pregnancy [35, 39–42]. Old age, living in mountainous region, being Islamic follower, and being smoker were identified as a risk factor for khat chewing during pregnancy [34, 43, 44].

Despite the considerable effect of substances use with birth defects and developmental disabilities, the current magnitude of caffeine and alcohol consumption, cigarette smoking and khat chewing during pregnancy were not studied in Ethiopia. Therefore, this study was aimed to estimate the prevalence of caffeine and alcohol consumption, khat chewing, and tobacco use during pregnancy and identify factors associated with excessive caffeine consumption.

## Materials and methods

### Study area

The study was conducted in *kebeles* covered by Butajira Rural Health Program (BRHP) which is found in Meskan and Mareko districts of Gurage Zone, SNNP region, South Central Ethiopia. BRHP is located 130 km South of Addis Ababa which is the capital city of Ethiopia. Butajira Rural Health Program is a Health Demographic Surveillance System (HDSS) Site for Addis Ababa University. It covers the selected *kebeles* of Meskan and Mareko district. The BRHP comprises one urban and nine rural *kebeles* (the smallest administrative unit in Ethiopia).

## Study design, period and population

A community based cross-sectional study was employed from April 12 to May 15, 2018. The source population of the study was all pregnant women living in *kebeles* covered by Butajira Rural Health Program. Pregnant women living in the study area were included in the study. None of the pregnant women was excluded.

## Sample size determination and sampling procedure

The sample size was determined by using single population proportion formula and based on the following assumptions; 5% of margin error (d) (except for alcohol use where d = 4.5%, and tobacco use where d = 1.6%) and with 57% expected prevalence of caffeine intake during pregnancy [45], 34% expected prevalence of alcohol consumption [18], 3% expected prevalence of cigarette smoking [24] and 40.7% expected prevalence of khat chewing [34]. After adding a 5% non-response rate and population correction, a final sample of 352 pregnant women were required.

A stratified sampling technique was used to select pregnant women as the accessibility of substances might vary based on residence and agro-ecological zones. First, the BRHP *kebeles* were stratified as urban and rural. Second, the rural *kebeles* were stratified as lowland, midland, and highland while the urban *kebele* is found in the midland agro-ecological zone. Third, the total number of pregnant women during the data collection period was obtained from the Butajira Rural Health Program database. A sampling frame was prepared for each stratum and the samples were assigned for each stratum proportional to the number of pregnant women. Finally, pregnant women were selected from each stratum by using a simple random sampling (SRS) technique.

## Data collection tool and procedure

We adapted a questionnaire from Caffeine Consumption Questionnaire-Revised (CCQ-R) [46] for caffeine consumption assessment, Alcohol Use Disorder Identification Test (AUDIT) [47] and EDHS 2016 (26) for alcohol consumption assessment, the Global Adult Tobacco Survey (GATS) [48] and EDHS 2016 (26) for tobacco use assessment and from published literatures [43] and EDHS 2016 (26) for khat chewing assessment.

Interviewer-administered face to face data collection method was employed to collect data from each pregnant woman. The overall data collection process was supervised by the two coordinators of Butajira Rural Health Program.

## Measurement of study variables

**Caffeine measurement.**   We conducted a repeated 24-hour recall was to assess the habitual intake of caffeine. To control days of the week's variation, data was collected on 2 non-consecutive days of the week, i.e. one from week days and one from weekend days. All days of the week were considered in the sample to make the selection representative.

We asked participants to report their last 24-h consumption of caffeine from different caffeinated beverages (coffee, tea, Coca-Cola, Pepsi cola, and energy drinks), caffeinated foods (e.g., chocolate, candy bars, and baked goods) but not from medicines. The data collectors asked respondents to show the serving size for each caffeinated item. If the material was not available at home, a picture of calibrated serving size was shown to the mother to help estimate the amount consumed. Then, the participants were asked to indicate how many of each size of the beverages consumed in the last 24-hr. The daily 24-hr recall consumption data were transformed in to a standard unit (ml). The data were multiplied by the content of caffeine per unit

of each caffeine source. The amount of caffeine from coffee (0.5309 mg/ml) was obtained from a study conducted in Ethiopia [49]. The caffeine concentration for tea (0.359 mg/ml) and coca cola (0.113mg/ml) was obtained from the International Food Information Council Foundation (IFICF) critical review on clarifying the controversies of caffeine and health [50].

To obtain the caffeine amount from coffee with milk, the portion of coffee in the drink (coffee with milk) was first estimated by conducting a pilot study before the actual data collection period. Based on the pilot study we conducted on 12 households, the estimated proportion of coffee in the drink of coffee served with milk was 0.7365. Then, the caffeine level was computed based on the caffeine level estimation in coffee as shown above.

The caffeine intake from each source was added to obtain the daily caffeine intake. The total level of caffeine consumption for two days was obtained by adding caffeine intake from each 24-h recall. The two days' consumption was divided by 2 to obtain the average caffeine intake of the pregnant woman per day. If the daily caffeine consumption was greater than or equal to 300mg, the woman labeled as excessive caffeine consumer.

**Alcohol consumption measurement.** The participants were asked to report their alcohol consumption during pregnancy using the 3 consumption questions from Alcohol Use Disorder Identification Test (AUDIT) [47]. If a woman reported as consuming at least one unit of alcohol from any sources (Tella, Teje, Areqe, Beer, Wine, and Distilled sprites) during the current pregnancy, she was labeled as alcohol consumer. If she consumed five or more alcohol drinks in one session (one sit) during the current pregnancy, she was labeled as a binge alcohol consumer.

**Cigarettes smoking assessment.** The pregnant woman was asked to respond to close-ended questions concerning cigarette smoking. First, the woman was asked a question to assess whether or not she smoked cigarette during the current pregnancy. If she responded "yes", the woman was labeled as a smoker and number of cigarettes smoked per day, awareness and source of information about the effects of cigarettes on the fetus, and secondary exposure at home, workplaces and public places were asked. If the pregnant woman exposed to tobacco at home during current pregnancy or work place in the last one month of the interview or public places in the last 7 days, she was labeled as a passive tobacco smoker.

**Khat chewing assessment.** The pregnant woman was asked to respond to a close-ended questions concerning khat chewing. First, the woman was asked a question to assess whether or not she chewed khat during the current pregnancy. If a woman responded "yes", she was labeled as a consumer and frequency of chewing, awareness, and source of information about the effects of khat chewing on the fetus were asked.

## Data quality assurance

To assure the quality of caffeine data, the locally available serving sizes of caffeinated products were calibrated and standardized before the data collection. In addition, two days training was given to the data collectors and supervisors to minimize an introduction of information bias. A pretest was conducted at neighboring kebele on 21 pregnant women before the actual data collection to check the consistency, any ambiguity in the language and to evaluate the skill of the interviewers. During pretest, new locally available serving sizes of caffeinated beverages were found and calibrated. Moreover, the questionnaire was modified based on the input from the pretest.

In addition, the consistency, completeness, and clarity of the data was checked by the data collectors before leaving the respondent and also checked by the supervisor and investigator on daily bases.

## Data management, analysis and presentation

The collected data was compiled, checked for any inconsistency and missed value, coded, and entered using Epi-data version 3.1 Software and exported into Stata 14 for data management and analysis. The data were cleaned for missing values by running frequencies, and crosstabs.

The normality of age and caffeine data was checked using histogram with normal curve, and Shapiro Wilk test p-value. For caffeine data, the minimum and maximum, and median with interquartile range were analyzed. Prevalence's were estimated for excess caffeine intake per day, alcohol consumption, khat chewing, passive smoking and overall substances use during pregnancy. Chi-square test statistics was run to test whether the proportion of excess caffeine consumption significantly differ across trimesters.

Principal Component Analysis was done to construct wealth index based on household data such as ownership of household including household fixed assets, type of house and its building materials, agricultural land ownership, animal ownership, source of drinking water, ownership and type of toilet facility, having domestic servant, and saving account. Assets owned by less than 5% or more than 95% of households were excluded from wealth index construction.

Bivariate and multivariate binary logistic regression was run to identify risk factors for excess daily caffeine intake among pregnant women. Those variables with p-value of less than 0.25 in bivariate logistic regression model were entered into the multivariate logistic regression model. Statistically significant association was declared at p value of less than 0.05.

## Ethical consideration

An ethical clearance as well as support letter was obtained from the Research Ethics Committee of School of Public Health (SPH), College of Health Sciences (CHS) of Addis Ababa University. An informed verbal consent was obtained from each respondent after explaining the information sheet of the study. The verbal consent was documented by the interviewer on the prepared consent form attached with the questionnaire. The obtained information from each respondent kept confidential.

# Results

A total 352 pregnant women approached for the study. Among these, we had a response rate of 96.9% (341 women). Table 1 shows the socio-demographic and economic characteristics of respondents. The majority of the respondents (79.2%) were rural residents. The median age of respondents in years was 28 (IQR = 6). The majority of respondents (61.9%) were within the age group of 25–34 years. In addition, the majority of respondents (83.9%) were Muslims and almost all (99.1%) were married.

## Prevalence and sources of caffeine consumption

The major sources of caffeine for pregnant women in the study are were coffee, coffee with milk, tea and Coca-Cola. Specifically, 88.9% of pregnant women (95% CI: 85.0, 91.8%) consumed coffee during the current pregnancy, 35.8% (95% CI: 30.8, 41.0%) consumed coffee with milk, 24.9% (95% CI: 20.6, 29.8%) consumed tea and 0.6% (95% CI: 0.1, 2.3%) consumed Coca-Cola. However, Pepsi cola, energy drinks and chocolates were not the sources of caffeine among pregnant women in the study area.

Almost all pregnant women (98.2%; 95% CI: 96.1%, 99.2%) consumed caffeine as estimated using the 2 days 24-hour recall average. Table 2 shows the median, range, and prevalence of excess caffeine intake among pregnant women by trimester of pregnancy. The daily caffeine intake among pregnant women ranges from 0.00–549.8 mg per day. The median daily caffeine

**Table 1. Socio-demographic and economic characteristics of pregnant women in Butajira, South Central Ethiopia, 2018.**

| Variables | Frequency | Percent (%) |
|---|---|---|
| **Residence** | | |
| Rural | 270 | 79.2 |
| Urban | 71 | 20.8 |
| **Climatic Zone** | | |
| Highland | 98 | 28.7 |
| Midland | 124 | 36.4 |
| Lowland | 119 | 34.9 |
| **Age** | | |
| 15–24 | 96 | 28.2 |
| 25–34 | 211 | 61.9 |
| 35 and above | 34 | 10.0 |
| **Religion** | | |
| Muslim | 286 | 83.9 |
| Orthodox Christian | 42 | 12.3 |
| Protestant | 13 | 3.8 |
| **Educational status** | | |
| No formal education | 148 | 43.4 |
| Primary Level | 150 | 44.0 |
| Secondary level and above | 43 | 12.6 |
| **Marital status** | | |
| Married | 338 | 99.1 |
| Separated/widowed | 3 | 0.9 |
| **Occupation** | | |
| House wife | 276 | 80.9 |
| Government Employee | 7 | 2.1 |
| Merchant | 45 | 13.2 |
| Others* | 13 | 3.8 |
| **Wealth status** | | |
| Poorest | 77 | 22.6 |
| Poor | 76 | 22.3 |
| Middle | 63 | 18.5 |
| Rich | 63 | 18.5 |
| Richest | 62 | 18.2 |

*Includes (farmer, daily laborer, tea/coffee sellers, free service, lumbering & hair making)

intake during pregnancy was 170.5 mg (IQR = 135.1). Across the sample, 17.6% (95% CI: 13.9%, 22.0%) of the pregnant women consumed 300 mg and above caffeine per day. The proportion of excess caffeine ($\geq$300mg/day) consumption was significantly different across the trimesters of pregnancy (p<0.001). The highest proportion of excess caffeine consumption was observed at the first trimester of pregnancy (50%; 95% CI: 27.2, 72.8). About 41.9% (95% CI: 36.8%, 47.3%) of the pregnant women consumed 200 mg and above caffeine per day.

## Prevalence of alcohol consumption during pregnancy

Table 3 shows the prevalence of alcohol consumption among pregnant women. From a total of 341 pregnant women, 11.1% (95% CI: 8.2%, 15.0%) of the pregnant women consumed alcohol in the last 3 months before current pregnancy and 10.0% (95% CI: 7.2%, 13.7%) consumed

**Table 2. Estimated caffeine intake among pregnant women based on the non-consecutive repeated 24 hours recall average, by trimester of pregnancy, Butajira, South Central Ethiopia, 2018.**

| Trimester of pregnancy | No. | Median Caffeine intake (IQR) mg/day | Range of caffeine intake mg/day | Prevalence excessive caffeine intake ($\geq$300mg/day*) percent (95% CI) |
|---|---|---|---|---|
| 1st Trimester | 20 | 298.5 (199.1) | 39.5–549.8 | 50.0 (27.2, 72.8) |
| 2nd Trimester | 133 | 169.9 (133.4) | 0.0–525.0 | 15.8 (10.0, 23.1) |
| 3rd Trimester | 188 | 165.6 (125.0) | 0.0–505.4 | 15.4 (10.6, 21.4) |
| Total | 341 | 170.5(135.1) | 0.0–549.8 | 17.6 (13.9, 22.0) |

IQR = Inter Quartile range; CI = Confidence Interval

*300mg/day the maximum daily caffeine intake limit for pregnant women

alcohol during the current pregnancy. Moreover, 4.4% (2.7, 7.2%) of pregnant women were binge drinkers (consumed five or more drinks at a single occasion). Among women who consumed alcohol during the current pregnancy, the majority (76.8%) consumed alcohol with a frequency of monthly or less than monthly, and almost all (94.1%) of them consumed 1–2 drinks on a typical day whenever they drink.

**Table 3. The prevalence of alcohol consumption among pregnant women in Butajira, South Central Ethiopia, 2018.**

| Variables | Frequency | Percent (95%CI) |
|---|---|---|
| **Woman ever consumed alcohol** | | |
| Yes | 45 | 13.2 (10.0, 17.2) |
| No | 296 | 86.8 (82.8, 90.0) |
| **Woman consumed alcohol 3 months before pregnancy** | | |
| Yes | 38 | 11.1 (8.2, 15.0) |
| No | 303 | 88.9 (85.0, 91.8) |
| **Alcohol intake during current pregnancy** | | |
| Yes | 34 | 10.0 (7.2, 13.7) |
| No | 307 | 90.0 (86.3, 92.8) |
| **Frequency of alcohol consumption (n = 34)** | | |
| Monthly or less | 26 | 76.8 |
| Two to four times a month | 7 | 20.6 |
| Two to three times a week | 1 | 2.9 |
| **Number of drinks on a typical day (n = 34)** | | |
| 1–2 | 32 | 94.1 |
| 3–4 | 2 | 5.9 |
| **Frequency of drinking five or more drinks in one occasion (n = 34)** | | |
| Never | 19 | 55.9 |
| Monthly | 3 | 8.8 |
| Less than monthly | 11 | 32.4 |
| Weekly | 1 | 2.9 |
| **Binge drinking (n = 341)** | | |
| Yes | 15 | 4.4 (2.7, 7.2) |
| No | 326 | 95.6 (92.8, 97.3) |
| **Obtained information to stop drinking alcohol (n = 34)** | | |
| Yes | 5 | 14.7 |
| No | 29 | 85.3 |

## Prevalence of khat chewing during pregnancy

Table 4 shows the prevalence, frequency and amount of khat chewing among pregnant women. A total of 122 (35.8%; 95% CI: 30.8, 41.0%) of pregnant women chewed khat during the current pregnancy. Among these, 36 (29.5%) chewed khat two to three times a week while 6.6% of them chewed khat more than four times a week.

## Prevalence of active and passive tobacco smoking during pregnancy

Table 5 shows the proportion of pregnant women exposed to tobacco smoke at home, work and public places. None of the pregnant women was active tobacco smokers. However, 9.7% (95% CI: 6.9, 13.3%) of pregnant women exposed to tobacco smoke at home. Of them, the majority (75.8%) of pregnant women exposed to tobacco smoke daily. Overall, 23.2% (95% CI: 19.0, 28.0) of pregnant women were passive smokers.

## Overall prevalence of substances use during pregnancy

Table 6 shows the summary of substances use during pregnancy. The overall prevalence of substance use during pregnancy is defined by exposure at least to one of the four substances i.e. caffeine more than or equal to 300 mg, alcohol intake or khat chewing or tobacco smoke during current pregnancy. Based on this, the overall substance use during the current pregnancy was 60.1% (95% CI: 54.8%, 65.2%).

## Factors associated with excess caffeine consumption

Table 7 shows the result of a multivariate logistic regression analysis fitted to identify risk factors for excess caffeine consumption. After adjustment for possible confounders such as maternal age, educational status, wealth status, gestational age, antenatal care, awareness about the effects of coffee/tea consumption on the fetus and khat chewing during pregnancy, we

**Table 4. The prevalence, frequency and amount of khat chewing among pregnant women in Butajira, South Central Ethiopia, 2018.**

| Variables | | Frequency | Percent (95%CI) |
|---|---|---|---|
| **Woman chewed khat before the current pregnancy** | | | |
| | Yes | 147 | 43.1 (37.9, 48.4) |
| | No | 194 | 56.9 (51.6, 62.1) |
| **Woman ever chewed khat during current pregnancy** | | | |
| | Yes | 122 | 35.8 (30.8, 41.0) |
| | No | 219 | 64.2 (59.0, 69.2) |
| **Frequency of khat chewing (n = 122)** | | | |
| | Monthly or less | 47 | 38.5 |
| | Two to four times a month | 30 | 24.6 |
| | Two to three times a week | 36 | 29.5 |
| | Four or more times a week | 8 | 6.6 |
| | Daily | 1 | 0.8 |
| **Woman chewed khat in the last 30 days of interview (n = 341)** | | | |
| | Yes | 115 | 33.7 (28.9, 38.9) |
| | No | 226 | 66.3 (61.1, 71.1) |
| **Informed to stop chewing (n = 122)** | | | |
| | Yes | 6 | 4.9 |
| | No | 116 | 95.1 |

**Table 5. The prevalence of passive smoking among pregnant women in Butajira, South Central Ethiopia, 2018.**

| | Variables | Frequency | Percent (95% CI) |
|---|---|---|---|
| **Exposure to smoking at home** | | | |
| | Yes | 33 | 9.7 (6.9, 13.3) |
| | No | 308 | 90.3 (86.7, 93.1) |
| **Frequency of another person smoked at home (n = 33)** | | | |
| | Daily | 25 | 75.8 |
| | Weekly | 4 | 12.1 |
| | Monthly | 2 | 6.1 |
| | Less than monthly | 2 | 6.1 |
| **Exposure to smoking at work places in the last 30 days** | | | |
| | Yes | 2 | 0.6 (0.1, 2.3) |
| | No | 339 | 99.4 (97.7, 99.9) |
| **Exposure to smoking at Public places in the last 7 days** | | | |
| | Yes | 52 | 15.2% (11.8, 19.5) |
| | No | 289 | 84.8% (80.5, 88.2) |
| **Overall passive smoking** | | | |
| | Yes | 79 | 23.2 (19.0, 28.0) |
| | No | 262 | 76.8 (72.0, 81.0) |

found that the poor, rich and richest wealth status and first trimester pregnancies were significantly associated with excess caffeine consumption among pregnant women. The odds of excessive caffeine consumption is approximately four times higher among pregnant women at poor wealth status compared to the odds among the pregnant women at the poorest wealth status (AOR = 3.63; 95% CI: 1.16, 11.32). Similarly, as compared to the odds of excessive caffeine consumption among the pregnant women at the poorest wealth status, the odd of excessive caffeine consumption was approximately four times higher among pregnant women at the rich wealth status (AOR = 3.74; 95% CI: (1.17, 11.88). In addition, the odds of excessive caffeine consumption is approximately four times higher among pregnant women at richest wealth status compared to the odds of excessive caffeine consumption among the pregnant women at the poorest wealth status (AOR = 3.66; 95% CI: 1.13, 11.88).

Moreover, the odds of excessive caffeine consumptions is four times higher among pregnant women at the first trimester of pregnancy compared to the odds among the pregnant women at third trimester (AOR = 4.04; 95% CI: 1.26, 13.05).

## Discussions

We conducted a community based cross sectional study to determine the prevalence of excess caffeine, alcohol consumption, khat chewing, and tobacco use during pregnancy and identify factors associated with excessive caffeine consumption. The study found that 17.6% of

**Table 6. The summary of substances uses among pregnant women in Butajira, South Central Ethiopia, 2018.**

| Variables | Affirmative response | Frequency | Percent (95% CI) |
|---|---|---|---|
| Excess caffeine intake | Yes | 60 | 17.6 (13.9, 22.0) |
| Alcohol intake | Yes | 34 | 10.0 (7.2, 13.7) |
| Khat chewing | Yes | 122 | 35.8 (30.8, 41.0) |
| Passive smokers | Yes | 79 | 23.2 (19.0, 28.0) |
| At least one substances use | Yes | 205 | 60.1 (54.8, 65.2) |

**Table 7. The multivariable logistic regression analysis to identify factors associated with excess caffeine consumption among pregnant women in Butajira, South Central Ethiopia, 2018.**

| Variables | Excess Caffeine Intake | | COR (95% CI) | +AOR (95% CI) |
|---|---|---|---|---|
| | No Count (%) | Yes Count (%) | | |
| **Age** | | | | |
| 15–24 | 85 (88.5) | 11 (11.5) | 1.00 | 1.00 |
| 25–34 | 177 (83.9) | 34 (16.1) | 1.48 (0.72, 15.36) | 0.88 (0.38, 2.04) |
| 35 and above | 19 (55.9) | 15 (44.1) | 6.10 (2.42, 15.36) *** | 2.98 (0.97, 9.14) |
| **Educational status** | | | | |
| No formal education | 111 (75.00) | 37 (25.00) | 2.06 (0.80, 5.26) | 0.85 (0.29, 2.48) |
| Primary | 133 (88.70) | 17 (11.30) | 0.79 (0.29, 2.14) | 0.39 (0.13, 1.16) |
| Secondary and above | 37 (86.00) | 6 (14.00) | 1.00 | 1.00 |
| **Wealth status** | | | | |
| Poorest | 72 (93.5) | 5 (6.5) | 1.00 | 1.00 |
| Poor | 62 (81.6) | 14 (18.4) | 3.25 (1.11, 9.54) * | 3.63 (1.16, 11.32) * |
| Middle | 50 (79.4) | 13 (20.6) | 3.74 (1.26, 11.17) * | 2.35 (0.72, 7.69) |
| Rich | 49 (77.8)) | 14 (22.2) | 4.11 (1.39, 12.16) * | 3.74 (1.17, 11.88) * |
| Richest | 48 (77.4) | 14 (22.6) | 4.20 (1.42, 12.42) ** | 3.66 (1.13, 11.88) * |
| **Trimester** | | | | |
| 1st | 10 (50.0) | 10 (50.0) | 5.48 (2.10, 14.34) ** | 4.54 (1.38, 15.00) * |
| 2nd | 112 (84.2) | 21 (15.8) | 1.03 (0.56, 1.89) | 0.82 (0.41, 1.64) |
| 3rd | 159 (84.6) | 29 (15.4) | 1.00 | 1.00 |
| **Antenatal care Follow up** | | | | |
| Yes | 256 (85.0) | 45 (15.0) | 1.00 | 1.00 |
| No | 25 (62.5) | 15 (37.5) | 3.41 (1.67, 6.97) *** | 2.25 (0.94, 5.36) |
| **Awareness on excess caffeine effect on the fetus** | | | | |
| Yes | 71 (92.2) | 6 (7.8) | 1.00 | 1.00 |
| No | 210 (79.5) | 54 (20.5) | 3.04 (1.26, 7.38) * | 2.36 (0.92, 6.05) |
| **Khat chewing** | | | | |
| Yes | 91 (74.6) | 31 (25.4) | 2.23 (1.27, 3.93) ** | 1.71 (0.90, 3.25 |
| No | 190 (86.8) | 29 (13.2) | 1.00 | 1.00 |

*p value <0.05

** p value <0.01

*** p value <0.001

**COR:** Confidence Interval, **COR:** Crude Odds ratio, **AOR:** Adjusted odds ratio

+Adjusted for maternal age, educational status, wealth status, gestational age, ANC, awareness about the effects of coffee/tea consumption on the fetus and khat chewing during pregnancy

pregnant women had a daily caffeine consumption more than or equal to 300 mg. In addition, one in ten and nearly four in ten pregnant women consume alcohol and chew khat during pregnancy respectively. We also found that two in ten pregnant women are passive tobacco smokers. Additionally, after adjustment for possible confounders, richest wealth status, and first trimester pregnancy were significantly associated with excess caffeine consumption among pregnant women.

The current study showed that the prevalence of excessive caffeine consumption (more than or equal to 300 mg per day) among pregnant women was 17.6%. The prevalence of caffeine consumption more than or equal to 200 mg per day among pregnant women was 41.9%. A great proportion (50.0%) of pregnant women in the first trimester of pregnancy consumed

excessive caffeine compared to the pregnant women at the second and third trimester of pregnancy. This finding is not in line with the previous literature. According a study in 2004, 96% of subjects decreased or quit drinking coffee during first trimester pregnancy [51]. The possible reason for this discrepancy and high prevalence of excess caffeine intake during first trimester of pregnancy needs further investigation.

Based on the existing literature, high levels of caffeine intake during pregnancy can result in miscarriage, low birth weight, growth restriction, stillbirth, and increases the risk of health problems in later life [52–59]. This implies that pregnant women were at risk of experiencing spontaneous abortion, stillbirth and low birth weight baby, which need an intervention.

This study found that 10.0% of pregnant women consumed alcohol during the current pregnancy. This figure is comparable to the global (9.8%) (16) and national (7.9%) prevalence of alcohol consumption among pregnant women (17). However, this finding was lower compared to a study from Ireland (60%), Belarus (47%), Denmark (46%), United Kingdom (41%), Russian Federation (37%) (20), South Eastern Nigeria (22.6%) (38), and Ethiopia at Bahirdar (34%) (18).

Maternal alcohol intake during pregnancy results in direct and indirect consequences on fetal development. Directly, alcohol readily crosses the placenta and blood-brain barriers and rapidly diffuses into any aqueous compartment of the body, such as the neurons or lipid membranes [60]. Exposure to alcohol during fetal development has been reported to reduce up to 12% of total brain weight, defined as microcephaly, due to decreased protein synthesis, which leads to decreased DNA translation [61]. Indirectly, alcohol induces maternal hypoxia, oxidative stress, and altered metabolism, affecting the growth and development of the fetus [62].

In addition, many alcoholics do not consume a balanced diet considering alcoholic beverages as part of their normal diet and acquire a certain number of calories from alcohol in substitution of calories from other nutrients. Moreover, alcohol consumption can interfere with the absorption of nutrients, impairing the quality and quantity of proper nutrient and energy intake, resulting in malnutrition especially of micronutrients such as vitamins, omega–3, folic acid, zinc, choline, iron, copper, and selenium [63]. When maternal nutritional status is compromised by alcohol the supply of essential nutrients are not available for the fetus; this can result in fetal abnormalities like Intrauterine Growth Restriction, Fetal Alcohol Spectrum Disorder [64], low birth weight (21), and small for gestational age (22). This indicated the need of alcohol consumption intervention program among pregnant women to prevent these adverse pregnancy outcomes.

In the current study the prevalence of khat chewing during the current pregnancy was 35.8% which is higher compared to the national khat chewing prevalence among the general population in Ethiopia (44). This difference might be due to the accessibility of khat in the current study area since khat grows as a cash crop in Butajira [65]. Moreover, low awareness on the harmful effects of substances use on the fetus as evidenced by this study might be the other possible reason. However figure is lower compared to a community based study finding in Yemen (34).This might be due socio-cultural variation.

Khat chewing during pregnancy may have different obstetric effects like low birth weight, stillbirths, impaired lactation, and embryo toxic as well as teratogenic properties. A study on rats in 1994 revealed that khat had retarded fetal growth and teratogenic effect and this developmental toxicity of khat is dose-related [66]. A case-control study conducted in 2015 at Bale Hospital of South East Ethiopia showed that maternal history of khat chewing was associated with low birth weight [67]. Another case-control study in 2017 obtained similar finding [68].

In addition, khat chewing during pregnancy associated with restrictive dietary behavior which results in Anemia. According to a study in 2013, the risk of anemia was 29% higher in the women who chewed khat daily than those who chewed sometimes or never did so [69].

These indicated that the need of khat chewing intervention program to prevent maternal anemia and adverse pregnancy outcomes.

In the current study, none of the pregnant women was active tobacco smokers. However, the prevalence of passive smoking was 23.2%. This figure is lower as compared to the findings of a study conducted in Shanghai, China (2016) where it was 34.8% [70]. Though the figure was lower, active or passive tobacco smoke exposure during pregnancy has adverse health effects on the fetus, as well as the mother. The adverse health effect of cigarette smoke on the fetus includes, an increased risk of strabismus in the offspring [71], clubfoot [72], low birth weight for gestational age (LBWGA), low birth weight, preterm births (28, 29), increased odds of elevated levels of antisocial behaviors during adolescence and adulthood, as well as violent and nonviolent outcomes [73], an increased risk of wheeze in children [74], and almost 3 times increased risk of congenital heart defects [75]. Moreover, tobacco smoke during pregnancy increases the prevalence of depressive symptoms during pregnancy [76]. This indicated the need of tobacco smoking intervention at home, work and public places to improve fetal, maternal and societal health.

The current study showed that the odds of excessive caffeine consumption was approximately four times higher among pregnant women at the poor, rich and richest wealth status compared to the odds of excessive caffeine consumption among pregnant women at the poorest wealth status. Likewise, the odd of excessive caffeine consumptions is four times higher among pregnant women at first trimester of pregnancy compared to the odds among the pregnant women at the third trimester of pregnancy. As women wealth status increases, the chance of buying coffee and consuming caffeinated beverages might increase. Due to this woman at the poor, rich and richest wealth status might consume excess caffeine compared to woman at the poorest wealth status. The reason why pregnant women at first trimester of pregnancy consume excess caffeine needs an investigation.

The findings of the study should be interpreted with the following strengths and limitations. Since we the study employed a random sampling methods, the findings could be generalizable to pregnant women living in the study area and similar settings. All days of the week were considered in order to control days of the week effect. In addition, non-consecutive 2 days repeated 24-hour recall which is the recommended method for the assessment of exposure within risk assessment processes was done to control with in person variation of caffeine intake.

However, the study has the following limitations. First, the level of caffeine concentration was obtained from previously conducted researches. However, the concentration of caffeine may vary based on the roasting and brewing process. Moreover, the strength of coffee was not considered as weak, medium and strong. Due to these reasons, the reported estimates of caffeine intake might be under or over estimated. Second, the study was conducted during the non-fasting season. In Ethiopia, coffee is traditionally consumed under coffee ceremony and the ceremony is a minimum of three times a day. In each ceremony, a person will take three cups of coffee [77]. Hence during fasting time, the number of ceremony decreases and this alter the daily coffee consumption. To the contrary, the frequency of coffee consumption increases during non-fasting season. Due to this reason our study might overestimate the prevalence of excessive caffeine consumption. Third, substances use such as alcohol and tobacco use are considered as non-religious and taboo in the study area. As a result, some respondents might not report consumption of alcohol and khat. These factors could underestimate the prevalence of alcohol and tobacco use among pregnant women. Fourth, there was no follow up of pregnant women. Due to this, the study could not assess exist of correlation between substance use and birth weight.

## Conclusions

In conclusion, the prevalence of excess caffeine consumption, khats chewing as well as passive tobacco smoking were high among pregnant women. However, the prevalence of alcohol consumption was comparable to the global and national prevalence. The richest wealth status and first trimester pregnancy were significantly identified risk factors associated with excess caffeine consumption among pregnant women. Therefore, interventional programs that addresses caffeine and alcohol consumption, khat chewing and Tobacco smoke exposures among pregnant women are needed. Moreover, further research is also needed to examine the effect of substance use on birth outcomes.

## Supporting information

**S1 Dataset.**
(DTA)

## Acknowledgments

First, we would like to acknowledge SPH, CHS of AAU for giving us an ethical clearance and writing a support letter to the study area. Second, our acknowledgment goes to Butajira Rural Health Program Staffs for provided us all the necessary information about the study area. At last but not least, we are grateful to all respondents for their voluntariness and participation.

## Author Contributions

**Conceptualization:** Alehegn Aderaw Alamneh, Bilal Shikur Endris, Seifu Hagos Gebreyesus.

**Data curation:** Alehegn Aderaw Alamneh.

**Formal analysis:** Alehegn Aderaw Alamneh.

**Investigation:** Alehegn Aderaw Alamneh.

**Methodology:** Alehegn Aderaw Alamneh, Bilal Shikur Endris, Seifu Hagos Gebreyesus.

**Project administration:** Alehegn Aderaw Alamneh.

**Resources:** Alehegn Aderaw Alamneh.

**Software:** Alehegn Aderaw Alamneh.

**Supervision:** Alehegn Aderaw Alamneh, Bilal Shikur Endris, Seifu Hagos Gebreyesus.

**Validation:** Alehegn Aderaw Alamneh.

**Visualization:** Alehegn Aderaw Alamneh.

**Writing – original draft:** Alehegn Aderaw Alamneh.

**Writing – review & editing:** Alehegn Aderaw Alamneh, Bilal Shikur Endris, Seifu Hagos Gebreyesus.

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
