## [Decision Letter · Decision Letter 0]

7 Feb 2020

PONE-D-19-29554

Caffeine, Alcohol, Khat, and Tobacco Use during Pregnancy in Butajira, South Central Ethiopia

PLOS ONE

Dear Mr. Alamneh,

Thank you for submitting your manuscript to PLOS ONE. After careful consideration, we feel that it has merit but does not fully meet PLOS ONE’s publication criteria as it currently stands. Therefore, we invite you to submit a revised version of the manuscript that addresses the points raised during the review process.

Please note that the reviewers express concerns and indicate that there are major issues regarding the methods, presentation and discussion of results that need to be addressed before acceptance. The criticisms range from requests of several clarifications in the methods to inclusion of new data.

We would appreciate receiving your revised manuscript by Mar 23 2020 11:59PM. To enhance the reproducibility of your results, we recommend that if applicable you deposit your laboratory protocols in protocols.io, where a protocol can be assigned its own identifier (DOI) such that it can be cited independently in the future. For instructions see: http://journals.plos.org/plosone/s/submission-guidelines#loc-laboratory-protocols

We look forward to receiving your revised manuscript.

Kind regards,

Yael Abreu-Villaça, Ph.D.

Academic Editor

PLOS ONE

Journal Requirements:

2. Please address the following:

- Please ensure you have thoroughly discussed any potential limitations of this study within the Discussion section. For example, the lack of follow-up with these participants does not allow for correlations to be made between substance use and birth weight.

- Please include additional information regarding the survey or questionnaire used in the study and ensure that you have provided sufficient details that others could replicate the analyses. For instance, if you developed a questionnaire as part of this study and it is not under a copyright more restrictive than CC-BY, please include a copy, in both the original language and English, as Supporting Information. In addition, please provide further details on the pre-testing of this questionnaire - i.e. how many participants were involved and from where were they recruited.

- Please provide additional details regarding participant consent. In the ethics statement in the Methods and online submission information, please ensure that you have specified how verbal consent was documented and witnessed.

- Please provide further details of your pilot study. This is mentioned in the text with no further information or reference to a published work.

Thank you for your attention to these queries.

Reviewers' comments:

Reviewer's Responses to Questions

**Comments to the Author**

1. Is the manuscript technically sound, and do the data support the conclusions?

Reviewer #1: No

Reviewer #2: Yes

Reviewer #3: Partly

2. Has the statistical analysis been performed appropriately and rigorously? 

Reviewer #1: Yes

Reviewer #2: Yes

Reviewer #3: Yes

3. Have the authors made all data underlying the findings in their manuscript fully available?

Reviewer #1: Yes

Reviewer #2: Yes

Reviewer #3: Yes

4. Is the manuscript presented in an intelligible fashion and written in standard English?

Reviewer #1: No

Reviewer #2: Yes

Reviewer #3: No

5. Review Comments to the Author

Reviewer #1: Dear author,

The following issues regarding Introduction, Methods, and Discussion sections must be addressed:

a. I think the study objective should be assess the prevalence of caffeine and alcohol consumption, khat chewing, and tobacco and predictors of caffeine consumption among pregnant pregnant women in South Central Ethiopia rather than "determine the magnitudes of caffeine and alcohol consumption, khat chewing, and tobacco use during pregnancy and identify factors associated with excess caffeine consumption".

b. The introduction is very long, some data are redundant. I suggest shortening and reviewing some paragraphs. Also, the last paragraph on page 5 would be better suited to the discussion.

c. The discussion should focus on the physiological and sociodemographic factors that may contribute to the increase in maternal caffeine intake. The study results should be confronted with others in the literature. However, the discussion has a strong focus on the adverse effects of maternal habits on the fetus and child that I consider to be more suitable for the introduction.

d. Was the design effect used for sample calculation?

e. Was the questionnaire validated?

f. It was not possible to quantify the concentration and the brand of caffeine products in this study. The author did not consider the strength of coffee as commonly consumed (weak, medium, or strong coffee). These limitations need to be included in the discussion.

g. Could the lack of information about some medications to underestimate the amount of caffeine ingested by pregnant women?

h. The literature report that social factors such as the family history of alcoholism, occupation of the head of the family, and higher parity are positively associated with high caffeine consumption. The author should comment on these limitations in the discussion.

i. Please define ANC follow up abbreviation (antenatal care).

j. Page 6: The sentence "The study was conducted in Butajira Rural Health Program (BRHP) site, which is found in Meskan and Mareko districts of Gurage Zone, SNNP region, South Central Ethiopia" is confusing. This needs to be more clearly described. Is the author saying the study was conducted in areas (kebeles) covered by BRHP?

k. Page 6: I think the sentences "Butajira Rural Health Program is a Health Demographic Surveillance System (HDSS) Site for Addis Ababa University" and "BRHP comprises one urban and nine rural kebeles (a kebele is the smallest administrative unit in Ethiopia)" are confusing. Does the Butajira Rural Health Program is a Health Demographic Surveillance System (HDSS) site? Does the BRHP cover selected kebeles of the Meskan and Mareko district? Please clarify it.

l. Page 7: "The source population of the study was all pregnant women living at BRHP". I’m not sure of what you mean "living at BRHP". Please, clarify it.

m. Page 7: "The list of pregnant women were obtained from the BRHP database". How about the recruitment process? How were they initially contacted? Was there a visit to the participant’s home? How about the exclusion or inclusion criteria?

n. Page 11: I think the sentence "After data collection, the data were entered using Epi-data version 3.1 software to avoid an introduction of error while data entry" is redundant. This information is described in the next paragraph.

o. Page 14: "However, Pepsi cola, energy drinks, and chocolates were not sources of caffeine among pregnant women in the study area pregnancy" I’m not sure of what you mean " sources of caffeine among pregnant women in the study area pregnancy ". Please, clarify it.

p. Page 22: The sentence “The possible reason for this high prevalence of excess caffeine intake during the first trimester of pregnancy needs further investigation” should be discussed. The literature report that in the first-trimester, behavioral changes occur aiming at enhancing personal care, and symptoms such as nausea or aversion to the coffee smell or taste can be quite common during pregnancy, reducing coffee consumption.

q. Page 25: The statement “ The current study showed that the odds of excessive caffeine consumption is approximately… at the richest wealth status compared to the odds among pregnant women at the poorest wealth status” seems no make much sense. Positive associations with poor women were seen. Besides, no significant association between middle income and coffee consumption were observed.

r. Page 25: “Due to this woman with richest wealth status might consume excess caffeine compared to woman with the poorest wealth status” and “As women wealth status increases, the chance of buying coffee and consuming caffeinated beverages might increase.” Poor people of rural areas do not consume coffee? Is coffee expensive in the studied areas? Does the studied area plant coffee?

s. Page 26: I’m not sure I understand this statement “Second, the study was conducted during the non-fasting season, and our report might overestimate the prevalence of excess caffeine consumption.” Is the author saying that women consume more caffeine because, in the fasting season, they can't do it? Please, clarify this statement.

t. The conclusion should be re-evaluated according to the considerations addressed in this review.

Reviewer #2: The manuscript’s issue is the excessive use of caffeine, and the consumption of alcohol, cigarette, and khat during pregnancy, and their adverse effects on the fetus's health. The theme is relevant and actual once it is a public health problem in many countries in the world. It is more critical in developing countries, in which many other socioeconomic risk factors can impact the pregnancy so, it is important to produce scientific information specifically from these countries.

The authors present global statistical data that strengthen the importance of the issue.

The abstract summarizes the arguments in an adequate way.

Method: the study area, study population, and sample size determination and sampling procedure are well described. A pilot study and a pretest were conducted and, a “two days training” was done to the data collectors and supervisors before the accomplishment of the study.

The authors considered that “the accessibility of substances might vary based on residence and agro-ecological zones”. It could be interesting to add information about how this accessibility could vary and which are the characteristics of these zones and their impact on drug use that could justify this division. Additionally, there are no comments about this point in the Results or Discussion sessions.

The results are well described, and the Discussion encompasses the main findings. The Conclusion is pertinent.

Reviewer #3: In the submitted manuscript the authors investigated the prevalence of caffeine, alcohol, khat, and tobacco use among 352 pregnant women in Ethiopia. They also investigated the main factors associated with the excess of caffeine intake among those women. The consumption of these substances during pregnancy may pose a risk to the health of mothers and newborns. Since information on the use of these substances during pregnancy in developing countries like Ethiopia is very scarce, this manuscript can bring new findings to the subject. However, to be published this manuscript needs some improvements.

MAJOR COMMENTS

The authors could have moved forward and, in addition to assess the use of caffeine, alcohol, khat, and tobacco during pregnancy, they could also have investigated the association between the use of these substances and adverse birth outcomes in the studied population. Although this association has been extensively studied, very few is known about this relationship under the socioeconomic conditions expected to be found in developing countries like Ethiopia.

English needs major revision.

MINOR COMMENTS

Page 3, line 54: The definition of "substance use" was retrieved from a website. I recommend that the authors search for a definition on a paper or book. In addition, I believe that authors meant to define substance abuse, not use.

Page 3, line 63: This submitted version still shows (ref), which I assumed as a missing reference.

The authors obtained the amount of caffeine intake from several sources through a two-24-hr recall survey, one in a weekday and another on a weekend day. Then, they calculated an average intake by summing the two-24-hr, and dividing by 2. Did authors test for differences in the caffeine intake between weekdays and weekend days for the different sources?

The authors found that the factors significantly associated with excess consumption of caffeine were the richest wealth status and the first trimester of pregnancy. However, they failed to describe that, compared with the poorest, the poor wealth status was also associated with excess of caffeine intake. In addition, crude odds ratios also suggested that age and antenatal care follow up may also be associated with the outcome. Although such associations were not statistically significant in the adjusted analyses, they were borderline and deserve to be mentioned in the results’ section.

6. PLOS authors have the option to publish the peer review history of their article (what does this mean?). If published, this will include your full peer review and any attached files.

Reviewer #1: No

Reviewer #2: No

Reviewer #3: Yes: Armando Meyer

---

## [Author Response · Author response to Decision Letter 0]

13 Mar 2020

PONE-D-19-29554

Caffeine, Alcohol, Khat, and Tobacco Use during Pregnancy in Butajira, South Central Ethiopia

PLOS ONE

Responses to the Academic Editor

Thank you for your valuable questions and suggestions. We have tried to respond the queries you raised on the method and limitation sections line by line as follow:

http://www.plosone.org/attachments/PLOSOne_formatting_sample_main_body.pdf and http://www.plosone.org/attachments/PLOSOne_formatting_sample_title_authors_affiliations.pdf

Response:

• Thank you! We have prepared our manuscript as per the PLOS ONE’s guideline. 

2. Please address the following:

- Please ensure you have thoroughly discussed any potential limitations of this study within the Discussion section. For example, the lack of follow-up with these participants does not allow for correlations to be made between substance use and birth weight.

Response: 

• Dear academic editor Thank you! We incorporated it as “There was no follow up of pregnant women. Due to this, the study could not assess exist of correlation between substance use and birth weight” (Check on discussion section, page 29, lines 468-470). 

- Please include additional information regarding the survey or questionnaire used in the study and ensure that you have provided sufficient details that others could replicate the analyses. For instance, if you developed a questionnaire as part of this study and it is not under a copyright more restrictive than CC-BY, please include a copy, in both the original language and English, as Supporting Information. In addition, please provide further details on the pre-testing of this questionnaire - i.e. how many participants were involved and from where were they recruited.

Response: 

Regarding the issues raised on the questionnaire, we did not develop a questionnaire. The questionnaire was adapted from Caffeine Consumption Questionnaire-Revised (CCQ-R), Alcohol Use Disorder Identification Test (AUDIT), and Global Adult Tobacco Survey (GATS), EDHS 2016 for caffeine, alcohol consumption, tobacco use and khat chewing assessment, respectively (Check on materials and methods section, page 8, lines 152-156). 

Concerning the pre-test, it was done on 21 pregnant women who recruited from out of the actual study area (Check on materials and methods section, page 11, lines 213-218). 

- Please provide additional details regarding participant consent. In the ethics statement in the Methods and online submission information, please ensure that you have specified how verbal consent was documented and witnessed.

Response: 

Regarding the verbal consent, the verbal consent was documented by the interviewer on the prepared consent form attached with the questionnaire (Check on materials and methods section, page 12, lines 245 & 246). 

- Please provide further details of your pilot study. This is mentioned in the text with no further information or reference to a published work. Thank you for your attention to these queries.

Response: 

Dear editor thank you! We conducted a pilot study before the actual data collection period to obtain the portion of coffee in the drink of coffee with milk. We recruited 12 households for the pilot study. Based on the pilot study we conducted, the estimated proportion of coffee in the drink of coffee served with milk was 0.7365 (Check on materials and methods section, page 9, lines 178-181). 

Response to Reviewer #1

Dear Reviewer,

Thank you for your valuable questions and suggestions. We have tried to address the queries you raised on the introduction, result, method and discussion sections as follow:

a. I think the study objective should be assess the prevalence of caffeine and alcohol consumption, khat chewing, and tobacco and predictors of caffeine consumption among pregnant women in South Central Ethiopia rather than "determine the magnitudes of caffeine and alcohol consumption, khat chewing, and tobacco use during pregnancy and identify factors associated with excess caffeine consumption".

Response: 

• Thank you for your suggestion! It was corrected as “To assess the prevalence of caffeine and alcohol consumption, khat chewing, and tobacco use during pregnancy and identify factors associated with excess caffeine consumption” (Check on the abstract section, page 2, line 27-29).

b. The introduction is very long, some data are redundant. I suggest shortening and reviewing some paragraphs. Also, the last paragraph on page 5 would be better suited to the discussion.

Response: 

• Thank you for your suggestion! To make the introduction more informative, it becomes somehow long as you said. If we shorten it, it will not be informative. That is why we obliged to narrate it as such. Nevertheless, we have tried to make it short. 

c. The discussion should focus on the physiological and sociodemographic factors that may contribute to the increase in maternal caffeine intake. The study results should be confronted with others in the literature. However, the discussion has a strong focus on the adverse effects of maternal habits on the fetus and child that I consider to be more suitable for the introduction.

Response: 

• Thank you for your insight! We have discussed those factors which are significantly associated with excess caffeine intake. In addition, we incorporated the effects of caffeine intake to show the implication of excess caffeine intake during pregnancy on the fetus (Check on discussion section, page 28, line 444-452).

d. Was the design effect used for sample calculation?

Response: 

• Because we have used stratified sampling technique, design effect was not considered during sample size determination.

e. Was the questionnaire validated?

Response: 

Dear reviewer, we did not develop a questionnaire. The questionnaire was adapted from Caffeine Consumption Questionnaire-Revised (CCQ-R), Alcohol Use Disorder Identification Test (AUDIT), and Global Adult Tobacco Survey (GATS), EDHS 2016 for caffeine, alcohol consumption, tobacco use and khat chewing assessment, respectively (Check on materials and methods section, page 8, lines 152-156). 

f. It was not possible to quantify the concentration and the brand of caffeine products in this study. The author did not consider the strength of coffee as commonly consumed (weak, medium, or strong coffee). These limitations need to be included in the discussion.

Response:

• Thank you! We incorporated it as a limitation (Check on discussion section, page 29, line 459-462).

g. Could the lack of information about some medications to underestimate the amount of caffeine ingested by pregnant women?

Response: 

• Yes, it underestimates the amount of caffeine consumption among pregnant women. Some drugs such as Aspirin contain caffeine as an ingredient. But, pregnant women mainly not took those drugs during pregnancy. 

h. The literature report that social factors such as the family history of alcoholism, occupation of the head of the family, and higher parity are positively associated with high caffeine consumption. The author should comment on these limitations in the discussion.

Response: 

• Dear reviewer, thank you for your insight. We have tried to search factors associated with excess caffeine consumption during pregnancy. But we could not find any study conducted on factors associated with excess caffeine consumption during pregnancy. 

i. Please define ANC follow up abbreviation (antenatal care). 

Response: 

• Thank you! ANC follow up corrected as “antenatal care follow up” (Check on Result section, page 23, Table 7, between line 261 & 262).

j. Page 6: The sentence "The study was conducted in Butajira Rural Health Program (BRHP) site, which is found in Meskan and Mareko districts of Gurage Zone, SNNP region, South Central Ethiopia" is confusing. This needs to be more clearly described. Is the author saying the study was conducted in areas (kebeles) covered by BRHP? 

Response: 

• Yes and it is corrected as “The study was conducted in Kebeles covered by Butajira Rural Health Program (BRHP) which is found in Meskan and Mareko districts of Gurage Zone, SNNP region, South Central Ethiopia.” (Check on materials and methods section, page 6, line 123-125).

k. Page 6: I think the sentences "Butajira Rural Health Program is a Health Demographic Surveillance System (HDSS) Site for Addis Ababa University" and "BRHP comprises one urban and nine rural kebeles (a kebele is the smallest administrative unit in Ethiopia)" are confusing. Does the Butajira Rural Health Program is a Health Demographic Surveillance System (HDSS) site? Does the BRHP cover selected kebeles of the Meskan and Mareko district? Please clarify it.

Responses: 

• Yes, Butajira Rural Health Program is a Health Demographic Surveillance System (HDSS) site (Check on materials and method, page 6 & 7, line 125-127).

• Yes, BRHP covers the selected kebeles of Meskan and Mareko district and clarified as suggested (Check on materials and method section, page 6 & 7, line 125-127).

l. Page 7: "The source population of the study was all pregnant women living at BRHP". I’m not sure of what you mean "living at BRHP". Please, clarify it.

Response: 

• Thank you! It is corrected as “The source population of the study was all pregnant women living at Butajira Rural Health Program” (Check on materials and method section, page 7, line 131 & 132).

m. Page 7: "The list of pregnant women was obtained from the BRHP database". How about the recruitment process? How they were initially contacted? Was there a visit to the participant’s home? How about the exclusion or inclusion criteria?

Responses:

How about the recruitment process?

• The total number of pregnant women during the data collection period was 466 as obtained from the Butajira Rural Health Program data base. Sampling frame was prepared for each stratum and the samples were assigned for each stratum proportional to the number of pregnant women. Then, study subjects were selected from each stratum by using simple random sampling (SRS) technique (Check on materials and method section, page 7, line 142-150). 

How they were initially contacted?

• The sampling frame was obtained from the Butajira Rural Health Program site. They continuously updated the database. Once we select the samples, the interview conducted at the respondents’ house (Check on materials and method section, page 7, line 145&146).

Was there a visit to the participant’s home? How about the exclusion or inclusion criteria?

• There was a visit to the participant’s home during the data collection time. 

• Pregnant women living in the study area were included in the study. None of the pregnant women was excluded (Check on materials and method, page 7, line 132 & 133). 

n. Page 11: I think the sentence "After data collection, the data were entered using Epi-data version 3.1 software to avoid an introduction of error while data entry" is redundant. This information is described in the next paragraph.

Response:

• Thank you! The statement “After data collection, the data were entered using Epi-data version 3.1 software to avoid an introduction of error while data entry” is removed as suggested (Check on materials and methods section, page 11, line 121). 

o. Page 14: "However, Pepsi cola, energy drinks, and chocolates were not sources of caffeine among pregnant women in the study area pregnancy" I’m not sure of what you mean "sources of caffeine among pregnant women in the study area pregnancy ". Please, clarify it.

Response:

• Thank you! It is corrected as “However, Pepsi cola, energy drinks and chocolates were not the sources of caffeine among pregnant women in the study area” (Check on result section, page 15, line 275 & 276). 

p. Page 22: The sentence “The possible reason for this high prevalence of excess caffeine intake during the first trimester of pregnancy needs further investigation” should be discussed. The literature report that in the first-trimester, behavioral changes occur aiming at enhancing personal care, and symptoms such as nausea or aversion to the coffee smell or taste can be quite common during pregnancy, reducing coffee consumption.

Response: 

• Thank you for your insight! We discussed as “This finding is not in line with the previous literature. According a study in 2004, 96% of subjects decreased or quit drinking coffee during first trimester pregnancy. The possible reason for this discrepancy and high prevalence of excess caffeine intake during first trimester of pregnancy needs further investigation” (Check on discussion section, page 25, line 381-383). 

q. Page 25: The statement “The current study showed that the odds of excessive caffeine consumption is approximately… at the richest wealth status compared to the odds among pregnant women at the poorest wealth status” seems no make much sense. Positive associations with poor women were seen. Besides, no significant association between middle income and coffee consumption were observed.

Response: 

• Thank you for your point of view! Though, there was a positive associations between excess caffeine consumption among poor women and excess caffeine consumption and no significant association between middle income and excess caffeine consumption were seen, the odds of excessive caffeine consumption is approximately four times higher among pregnant women at the richest wealth status as compared to the odds among pregnant women at the poorest wealth status. That is why we narrate and discussed it as such (Check on result section, page 22, line 342-347). 

r. Page 25: “Due to this woman with richest wealth status might consume excess caffeine compared to woman with the poorest wealth status” and “As women wealth status increases, the chance of buying coffee and consuming caffeinated beverages might increase.” Poor people of rural areas do not consume coffee? Is coffee expensive in the studied areas? Does the studied area plant coffee? 

Responses:

We are not saying that poor people of rural areas do not consume coffee. 

Yes, coffee is expensive in the study area. It is around 3 dollar per kg. 

The studied area does not plant coffee. 

s. Page 26: I’m not sure I understand this statement “Second, the study was conducted during the non-fasting season, and our report might overestimate the prevalence of excess caffeine consumption.” Is the author saying that women consume more caffeine because, in the fasting season, they can't do it? Please, clarify this statement.

Responses: 

• They can drink it during the fasting season. But, the authors want to say that the frequency of drinking coffee per day becomes decrease during the fasting season and increase during the non-fasting season. Due to this the intake of excess caffeine consumption may underestimated during the fasting period. While it may be overestimated during the non-fasting period (Check on discussion section, page 29, line 463-465). 

Response to Reviewer #2

Dear Reviewer,

Thank you for your valuable suggestions. We have tried to address the queries you raised on the method section as follow:

The manuscript’s issue is the excessive use of caffeine and the consumption of alcohol, cigarette, and khat during pregnancy, and their adverse effects on the fetus's health. The theme is relevant and actual once it is a public health problem in many countries in the world. It is more critical in developing countries, in which many other socioeconomic risk factors can impact the pregnancy so, it is important to produce scientific information specifically from these countries. The authors present global statistical data that strengthen the importance of the issue.

Response:

• Thank you!

• The abstract summarizes the arguments in an adequate way.

Method: the study area, study population, and sample size determination and sampling procedure are well described. A pilot study and a pretest were conducted and, a “two days training” was done to the data collectors and supervisors before the accomplishment of the study.

Response:

• Thank you!

• The authors considered that “the accessibility of substances might vary based on residence and agro-ecological zones”. It could be interesting to add information about how this accessibility could vary and which are the characteristics of these zones and their impact on drug use that could justify this division. Additionally, there are no comments about this point in the Results or Discussion sessions.

Response: 

• Dear reviewer thank you for your comment! The accessibility of substances at rural and urban areas may not be the same. For instance: Tobacco may not be accessible at rural areas as such to the urban areas. In addition, Khat grows mainly in highland than lowlands. Due to these reasons, the accessibility of substances use may vary based on residence and agro ecological zones (Check on materials and method, page 7, line 142 & 143). 

• The results are well described, and the Discussion encompasses the main findings. The Conclusion is pertinent.

Response:

• Thank you!

Response to Reviewer #3

Reviewer #3 comments

In the submitted manuscript the authors investigated the prevalence of caffeine, alcohol, khat, and tobacco use among 352 pregnant women in Ethiopia. They also investigated the main factors associated with the excess of caffeine intake among those women. The consumption of these substances during pregnancy may pose a risk to the health of mothers and newborns. Since information on the use of these substances during pregnancy in developing countries like Ethiopia is very scarce, this manuscript can bring new findings to the subject. However, to be published this manuscript needs some improvements. 

Dear Reviewer,

Thank you for your valuable questions and suggestions. We have tried to address the major and minor comments you raised as follow:

Response to Major Comments:

The authors could have moved forward and, in addition to assess the use of caffeine, alcohol, khat, and tobacco during pregnancy, they could also have investigated the association between the use of these substances and adverse birth outcomes in the studied population. Although this association has been extensively studied, very few is known about this relationship under the socioeconomic conditions expected to be found in developing countries like Ethiopia.

Response: 

• Thank you! We appreciate your point of view. In fact we did not assess the data related with adverse pregnancy outcomes. Due to this we cannot investigate any association between the use of these substances and adverse birth outcomes in the studied population. We mentioned this as a limitation in the discussion section (Check on discussion section, page 29, line 468-470). 

English needs major revision.

Response: 

• Thank you! We have revised the entire document.

Response to Minor comments

Page 3, line 54: The definition of "substance use" was retrieved from a website. I recommend that the authors search for a definition on a paper or book. In addition, I believe that authors meant to define substance abuse, not use.

Response: 

• Thank you for your recommendation. It would be nice if the source is from paper or book. But, we could not found any definition from book or paper. It is a medical dictionary definition, thus we decided to use as it is. 

• The authors mean to define substance use. Because, during pregnancy even substances use has an effect on the developing fetus. 

Page 3, line 63: This submitted version still shows (ref), which I assumed as a missing reference.

Response: 

• Thank you! We incorporated the missed reference (Check on introduction section, page 4 line 64). 

The authors obtained the amount of caffeine intake from several sources through a two-24-hr recall survey, one in a weekday and another on a weekend day. Then, they calculated an average intake by summing the two-24-hr, and dividing by 2. Did authors test for differences in the caffeine intake between weekdays and weekend days for the different sources?

Response: 

• We did not test for differences in the caffeine intake between weekdays and weekend days for the different sources. 

The authors found that the factors significantly associated with excess consumption of caffeine were the richest wealth status and the first trimester of pregnancy. However, they failed to describe that, compared with the poorest; the poor wealth status was also associated with excess of caffeine intake. In addition, crude odds ratios also suggested that age and antenatal care follow up may also be associated with the outcome. Although such associations were not statistically significant in the adjusted analyses, they were borderline and deserve to be mentioned in the results’ section.

Response: 

• Thank you! We described it as you suggested (Check on result section, page 22, lines 345-347).

---

## [Decision Letter · Decision Letter 1]

3 Apr 2020

PONE-D-19-29554R1

Caffeine, Alcohol, Khat, and Tobacco Use during Pregnancy in Butajira, South Central Ethiopia

PLOS ONE

Dear Mr. Alamneh,

Thank you for submitting your manuscript to PLOS ONE. After careful consideration, we feel that it has merit but does not fully meet PLOS ONE’s publication criteria as it currently stands. Therefore, we invite you to submit a revised version of the manuscript that addresses the points raised during the review process.

Please note that one, as detailed below, of the reviewers still has concerns regarding your statistical analysis and, as a consequence,  the results presentation and interpretation.

Please also modify the materials and method section, page 7, line 131 & 132 to make it clear that the population of the study was "all pregnant women living in kebeles covered by BRHP".

We would appreciate receiving your revised manuscript by May 18 2020 11:59PM. To enhance the reproducibility of your results, we recommend that if applicable you deposit your laboratory protocols in protocols.io, where a protocol can be assigned its own identifier (DOI) such that it can be cited independently in the future. For instructions see: http://journals.plos.org/plosone/s/submission-guidelines#loc-laboratory-protocols

We look forward to receiving your revised manuscript.

Kind regards,

Yael Abreu-Villaça, Ph.D.

Academic Editor

PLOS ONE

Reviewers' comments:

Reviewer's Responses to Questions

**Comments to the Author**

1. If the authors have adequately addressed your comments raised in a previous round of review and you feel that this manuscript is now acceptable for publication, you may indicate that here to bypass the “Comments to the Author” section, enter your conflict of interest statement in the “Confidential to Editor” section, and submit your "Accept" recommendation.

Reviewer #1: (No Response)

Reviewer #3: All comments have been addressed

2. Is the manuscript technically sound, and do the data support the conclusions?

Reviewer #1: Partly

Reviewer #3: Partly

3. Has the statistical analysis been performed appropriately and rigorously? 

Reviewer #1: Yes

Reviewer #3: Yes

4. Have the authors made all data underlying the findings in their manuscript fully available?

Reviewer #1: (No Response)

Reviewer #3: Yes

5. Is the manuscript presented in an intelligible fashion and written in standard English?

Reviewer #1: No

Reviewer #3: Yes

6. Review Comments to the Author

Reviewer #1: Dear author,

The vast majority of issues were adequately answered. However, there are some issues described below that must be taken into account in the interpretation of some results:

1) Page 25: “The current study showed that the odds of excessive caffeine consumption is approximately… at the richest wealth status compared to the odds among pregnant women at the poorest wealth status”. According to manuscript Table 7, the adjusted OR for excessive caffeine consumption according to wealth status was:

Poor woman vs poorest one (OR: 3.63, 95%CI: 1.16, 11.32)

Rich woman vs poorest one (OR: 3.74, 95%CI: 1.17, 11.88)

Richest woman vs. poorest one (OR: 3.66, 95%CI: 1.13, 11.88)

Looking at the CIs, if the author hypothetically conducts the same study with a different sample, the true value (OR) of excessive intake of caffeine for women poor in relation to the poorest may be between 1.16 and 11.32. For rich women compared to the poorest one, the true odds may be between 1.17 and 11.88, and for the richest women than the poorest woman, this true value might be between 1.13 and 11.88.

Regarding the issues raised above, is the Odds ratio from 3.63 different from 3.66 and different from 3.74?

Besides, an OR of 2 means there is a 100% increase in the odds of an outcome with a given "exposure". In relation to the paper results, in terms of magnitude, is an increase of 263% different from 274% and different from 266%? Please explain why only the result for the richest woman group compared to the poorest ones were highlighted and discussed.

2) Page 26: “...the authors want to say that the frequency of drinking coffee per day decreases during the fasting season and increases during the non-fasting season.”

During the fasting period, the types of food and drinks and their frequencies are changed (Masood et al. 2018). But I did not find any literature that states that religious fasting alters food patterns after fasting. Please, include references that support your statement.

Masood, S. N., Saeed, S., Lakho, N., Masood, Y., Ahmedani, M. Y., & Shera, A. S. (2018). Pre-Ramadan health seeking behavior, fasting trends, eating pattern and sleep cycle in pregnant women at a tertiary care institution of Pakistan. Pakistan journal of medical sciences, 34(6), 1326.

Reviewer #3: I have carefully reviewed my comments and suggestions sent to the authors. I'm satisfied with the authors' answers.

7. PLOS authors have the option to publish the peer review history of their article (what does this mean?). If published, this will include your full peer review and any attached files.

Reviewer #1: No

Reviewer #3: Yes: Armando Meyer, M.P.H, Ph.D

Professor of Occupational and Environmental Health

Public Health Institute

Federal University of Rio de Janeiro

ORCID: https://orcid.org/0000-0002-5258-8016

---

## [Author Response · Author response to Decision Letter 1]

14 Apr 2020

Response to the Academic Editor

We have tried to address the points raised on result presentation and interpretation during the review process line by line.

Comment 1: 

Please modify the materials and method section, page 7, line 131 & 132 to make it clear that the population of the study was "all pregnant women living in kebeles covered by BRHP".

Response:

Thank you for your suggestion! We have modified it as “The source population of the study was all pregnant women living in kebeles covered by Butajira Rural Health Program” (Check on Material and Method section, page 7, line 139 & 140). 

Response to Reviewer #1

Dear Reviewer,

Thank you for your invaluable concern. We have addressed the raised concerns on the result presentation and interpretation sections as follow:

1) Page 25: “The current study showed that the odd of excessive caffeine consumption is approximately… at the richest wealth status compared to the odds among pregnant women at the poorest wealth status”. According to manuscript Table 7, the adjusted OR for excessive caffeine consumption according to wealth status was:

Poor woman vs poorest one (OR: 3.63, 95%CI: 1.16, 11.32)

Rich woman vs poorest one (OR: 3.74, 95%CI: 1.17, 11.88)

Richest woman vs. poorest one (OR: 3.66, 95%CI: 1.13, 11.88)

Looking at the CIs, if the author hypothetically conducts the same study with a different sample, the true value (OR) of excessive intake of caffeine for poor women in relation to the poorest, may be between 1.16 and 11.32. For rich women compared to the poorest one, the true odds may be between 1.17 and 11.88, and for the richest women than the poorest woman, this true value might be between 1.13 and 11.88.

Regarding the issues raised above, is the Odds ratio from 3.63 different from 3.66 and different from 3.74? Besides, an OR of 2 means there is a 100% increase in the odds of an outcome with a given "exposure". In relation to the paper results, in terms of magnitude, is an increase of 263% different from 274% and different from 266%? Please explain why only the result for the richest woman group compared to the poorest ones were highlighted and discussed.

 Response: 

Dear reviewer, we really thank you for your insight! We have admitted your concern. Thus, we have described the odds ratio of excessive caffeine consumption among women in the poor wealth quintile as compared to the women in the poorest wealth quintile (Check on result section, page 23, lines 360-369). 

In addition, we have revised the discussion section (Check on discussion section, page 29, lines 464-466 and lines 470 & 471). 

2) Page 26: “...the authors want to say that the frequency of drinking coffee per day decreases during the fasting season and increases during the non-fasting season.”

During the fasting period, the types of food and drinks and their frequencies are changed (Masood et al. 2018). But I did not find any literature that states that religious fasting alters food patterns after fasting. Please, include references that support your statement.

Masood, S. N., Saeed, S., Lakho, N., Masood, Y., Ahmedani, M. Y., & Shera, A. S. (2018). Pre-Ramadan health seeking behavior, fasting trends, eating pattern and sleep cycle in pregnant women at a tertiary care institution of Pakistan. Pakistan journal of medical sciences, 34(6), 1326.

Response: 

Dear reviewer, sure, the authors want to say that the frequency of drinking coffee per day decreases during the fasting season and increases during the non-fasting season. Because, in Ethiopia, coffee is traditionally consumed under coffee ceremony and the ceremony is a minimum of three times a day. In each ceremony, a person will take three cups of coffee (Yoseph, 2013). Hence during fasting time, the number of ceremony decreases and this alter the daily coffee consumption. To the contrary, the frequency of coffee consumption increases during non-fasting season. Due to this reason our study might overestimate the prevalence of excessive caffeine consumption (Check on discussion section, page 30, lines 484-490). 

Response to Reviewer #3: 

Comment: I have carefully reviewed my comments and suggestions sent to the authors. I'm satisfied with the authors' answers.

Response: Dear reviewer, we are thankful for your invaluable contribution in the improvement of this manuscript.

---

## [Decision Letter · Decision Letter 2]

21 Apr 2020

Caffeine, Alcohol, Khat, and Tobacco Use during Pregnancy in Butajira, South Central Ethiopia

PONE-D-19-29554R2

Dear Dr. Alamneh,

We are pleased to inform you that your manuscript has been judged scientifically suitable for publication and will be formally accepted for publication once it complies with all outstanding technical requirements.

With kind regards,

Yael Abreu-Villaça, Ph.D.

Academic Editor

PLOS ONE

Additional Editor Comments (optional):

Reviewers' comments:

Reviewer's Responses to Questions

**Comments to the Author**

1. If the authors have adequately addressed your comments raised in a previous round of review and you feel that this manuscript is now acceptable for publication, you may indicate that here to bypass the “Comments to the Author” section, enter your conflict of interest statement in the “Confidential to Editor” section, and submit your "Accept" recommendation.

Reviewer #1: All comments have been addressed

2. Is the manuscript technically sound, and do the data support the conclusions?

Reviewer #1: Yes

3. Has the statistical analysis been performed appropriately and rigorously? 

Reviewer #1: Yes

4. Have the authors made all data underlying the findings in their manuscript fully available?

Reviewer #1: (No Response)

5. Is the manuscript presented in an intelligible fashion and written in standard English?

Reviewer #1: Yes

6. Review Comments to the Author

Reviewer #1: Dear author,

I am fully satisfied with the corrections. The presentation of the data is clear, and the overall quality of the manuscript seems adequate for publication in the PLOS ONE.

7. PLOS authors have the option to publish the peer review history of their article (what does this mean?). If published, this will include your full peer review and any attached files.

Reviewer #1: No

---

## [Editor Report · Acceptance letter]

29 Apr 2020

PONE-D-19-29554R2 

Caffeine, Alcohol, Khat, and Tobacco Use during Pregnancy in Butajira, South Central Ethiopia 

Dear Dr. Alamneh:

I am pleased to inform you that your manuscript has been deemed suitable for publication in PLOS ONE. Congratulations! Your manuscript is now with our production department. 

With kind regards,

on behalf of

Prof. Dr. Yael Abreu-Villaça 

Academic Editor

PLOS ONE